## Using electronic health records to develop and validate a machine-learning tool to predict type 2 diabetes outcomes: a study protocol

Ana Luisa Neves ![ORCID] ,[1,2] Pedro Pereira Rodrigues,[2] Abdulrahim Mulla,[3] Ben Glampson,[3] Tony Willis,[4] Ara Darzi,[1] Erik Mayer[1]

[1]NIHR Imperial Patient Safety Translational Research Centre, Imperial College London, London, UK
[2]Center for Health Technology and Services Research, Faculty of Medicine, University of Porto, Porto, Portugal
[3]Imperial College Healthcare NHS Trust, London, UK
[4]North West London Diabetes Transformation Programme, North West London Health and Care Partnership, London, UK

**Correspondence to**
Dr Ana Luisa Neves;
ana.luisa.neves14@imperial.ac.uk

## ABSTRACT

**Introduction** Type 2 diabetes mellitus (T2DM) is a major cause of blindness, kidney failure, myocardial infarction, stroke and lower limb amputation. We are still unable, however, to accurately predict or identify which patients are at a higher risk of deterioration. Most risk stratification tools do not account for novel factors such as sociodemographic determinants, self-management ability or access to healthcare. Additionally, most tools are based in clinical trials, with limited external generalisability.

**Objective** The aim of this work is to design and validate a machine learning-based tool to identify patients with T2DM at high risk of clinical deterioration, based on a comprehensive set of patient-level characteristics retrieved from a population health linked dataset.

**Sample and design** Retrospective cohort study of patients with diagnosis of T2DM on 1 January 2015, with a 5-year follow-up. Anonymised electronic healthcare records from the Whole System Integrated Care (WSIC) database will be used.

**Preliminary outcomes** Outcome variables of clinical deterioration will include retinopathy, chronic renal disease, myocardial infarction, stroke, peripheral arterial disease or death. Predictor variables will include sociodemographic and geographic data, patients' ability to self-manage disease, clinical and metabolic parameters and healthcare service usage. Prognostic models will be defined using multidependence Bayesian networks. The derivation cohort, comprising 80% of the patients, will be used to define the prognostic models. Model parameters will be internally validated by comparing the area under the receiver operating characteristic curve in the derivation cohort with those calculated from a leave-one-out and a 10 times twofold cross-validation.

**Ethics and dissemination** The study has received approvals from the Information Governance Committee at the WSIC. Results will be made available to people with T2DM, their caregivers, the funders, diabetes care societies and other researchers.

### Strengths and limitations of this study

► This study will develop and validate a machine learning-based tool to identify patients with type 2 diabetes mellitus (T2DM) at high risk of clinical deterioration, incorporating a comprehensive set of relevant variables (sociodemographic, geographic, clinical characteristics, patient self-management ability and healthcare service utilisation), which are often neglected in traditional risk scoring systems.

► Longitudinal, real-world patient data will be used, capitalising on linked electronic health records including data from primary, secondary, social and mental healthcare.

► The tool will be based in Bayesian Networks, an optimal method to conduct individual-level risk estimation, and easily transform the associations between variables into decision models.

► After validation, this tool has considerable potential to contribute to the decision-making process at patient level for this population, offer guidance to define pathways of care and to allocate economic and personnel resources.

► Limitations of the study include the potential lack of accuracy of diagnosis of T2DM and potentially related risk factors and/or clinical deterioration outcomes; however, being part of the quality and outcomes framework, T2DM is an area where healthcare professionals are particularly incentivised to keep information updated. Another limitation refers to the external validity of the tool, and therefore future work should consider replication in other populations.

## INTRODUCTION

Type 2 diabetes mellitus (T2DM) is one of the most common non-communicable diseases— and prevalence is progressively rising. There are currently 366 million people affected worldwide, and the total number is expected to increase to 552 million by 2030.[1] Uncontrolled diabetes is a well-recognised cause of blindness due to retinal damage, kidney failure and lower limb amputation and is associated with a threefold risk of cardiovascular disease.[2 3] Stemming from its high prevalence, morbidity and mortality, the management of people with T2DM implies important social

and financial costs, with the cost burden to healthcare systems increasing every year.[4]

In order to prevent or delay the onset of T2DM complications, it is critical to offer personalised care, which includes knowledge of which patients are at a higher risk of clinical deterioration.[5] While risk prediction models have considerable potential to contribute to the decision-making process at patient level, they can also offer guidance to define pathways of care, and to allocate economic and personnel resources. Although several T2DM risk scores were developed based on various regression models, some limitations need to be considered.

Risk prediction models are typically multivariate and combine several factors, but these tend to be mostly a priori known clinical risk factors. However, a large body of literature shows significant links between social and environmental factors and adverse health events,[6 7] and suggests that including these variables could likely improve the accuracy of risk prediction models.[8 9] Constructs comprise a range of socioeconomical influences on the individual, including but not limited to education, economic status and access to medical care.[10] Access to healthcare resources, in particular, may be unequal based on the patient's ethnicity, place of residence, socioeconomic status and education.[10] Additionally, patient activation (ie, self-confidence and ability to self-manage) has been associated with self-management behaviours, Hemoglobin A1c (HbA1c) knowledge and HbA1c testing frequency,[11 12] which may contribute to better achievement of glycaemic targets and therefore a lower risk of clinical deterioration. Despite the recognised importance of these factors, a systematic review evaluating the various risk prediction models for T2DM found they were seldom included. Only a few models included ethnicity (23%, n=10), social deprivation (5%, n=2) or education level (2.4%, n=1).[13] Previous literature also highlights a growing concern that the majority of risk prediction models are based on a suboptimal selection of the cohort,[13] raising concerns about the external validity and generalisability of the results. In this context, the use of real-world data from electronic health records (EHRs) can overcome many of the limitations of artificially selected cohorts. Additionally, a systematic review by Mahmoudi *et al* assessed the use of EHRs in the development and validation of risk prediction models and found that on average, models using EHRs data show better predictive performance.[9] However, the authors highlight that most of the models did not account for important socioeconomic features and lacked an adequate assessment of both clinical relevance and implementation.[9]

Additionally, the recent investments in EHR and their increasing use in healthcare have provided new opportunities to apply machine-learning methods.[14] A study comparing the prediction accuracies obtained by conventional statistical regression methods and machine-learning methods, in the context of T2DM, showed higher classification accuracies for machine-learning models.[15] In this context, the use of Bayesian Networks (BNs) is a widely used approach using probabilistic graphical models that represent a set of variables and their conditional dependencies, allowing the identification of relationships that may highlight causality.[16 17] These relationships are represented by a graphical structure, whereas the quantitative dependencies between individual variables are expressed as a conditional probability.[18] Over the last years, BNs have been extensively used to model diagnosis, risk assessment and disease prediction in the context of cardiovascular diseases,[19–21] but have not yet been much explored in the context of T2DM. Recently, a few studies developed promising approaches applying deep learning neural networks and comparative machine-learning approaches for predicting T2DM.[22–24]

In this work, we will use linked patient-level EHRs and BNs to analyse the relationship between a comprehensive set of patients' characteristics (including sociodemographic, geographic, clinical, patient activation and healthcare service utilisation) and clinical deterioration. Specifically, we will design and validate a tool able to differentiate patients with T2DM based on their probability of facing clinical deterioration, therefore allowing a strategic approach in what concerns preventive and therapeutic strategies.

## METHODS AND ANALYSIS
### Study design
A retrospective cohort study of patients with diagnosis of T2DM on 1 January 2015, with a 5-year follow-up. Patients with T2DM were identified by the corresponding Read codes (Quality and Outcomes Framework (QOF) business rules V.27).

### Data source and data management
Anonymised EHRs were accessed in the Whole System Integrated Care (WSIC) database.[25] Over 360 General Practice (GP) surgeries, 10 acute and specialist hospitals, 8 social care organisations and 2 mental health trusts and 2 community health trusts contribute to WSIC, which covers over 2.4 million patients in North West London, representing 30.0% of the London population.[25] Data from primary, secondary, community, social and mental healthcare are linked at patient level. The data used in the study are managed as part of WSIC systems, run and managed by the National Health System (NHS) and used for both direct care and approved research. This system pulls information from healthcare provider systems (local EHRs) under data controllers using a common model. Data are held in a secure NHS-managed environment. As our approach is data-driven (ie, we are using existing data to generate and validate new models, rather than finding evidence of a particular hypothesis), no specific sample size calculations were applicable, and therefore not performed.

### Study variables
Outcome variables of clinical deteriorations will include coded diagnosis of (1) retinopathy; (2) chronic renal

disease; (3) myocardial infarction; (4) stroke; (5) peripheral arterial disease or (6) death. Outcomes will be defined by relevant clinical codes (Read codes) for diagnoses.

Predictor variables included (1) sociodemographic and geographic data; (2) patients' ability to self-manage disease; (3) clinical and metabolic parameters and (4) healthcare service usage.

Sociodemographic information will be extracted at baseline, including age, gender, ethnicity, education, Index of Multiple Deprivation (IMD) and geographic location. The IMD is the official measure of relative deprivation in England and is part of a suite of outputs that form the Indices of Deprivation.[26] It follows an established methodological framework in broadly defining deprivation to encompass a wide range of an individual's living conditions. This is an overall measure of multiple deprivation experienced by people living in an area and is calculated for every lower-layer super output area or neighbourhood, in England.[26] Geographic location will be extracted as the first part of the postcode.[27]

Information on patient self-management ability will be extracted as Patient Activation Measure (PAM) values. The PAM is a validated tool of 13 questions, and it was delivered in its totality, without any edits or changes to the validated version.[26] Answers were weighted and combined to provide a score on a scale from 0 to 100.[28] PAM scores between 1 and 99 will be considered to be valid responses, and allow the categorisation into one of four predefined levels, ranging from 1 (patients who do not actively contribute to their healthcare) to 4 (patients who are proactive in managing their health and engage in healthy behaviours).[28] A score of less than 47.0 places a patient in level 1, 47.1–55.1 level 2, 55.2–72.4 level 3 and more than 72.5 in level 4.[28]

Clinical and metabolic factors extracted at baseline will include T2DM-related variables, such as HbA1c levels (%), diabetes treatment method (categorised as 'diet only', 'oral treatment' or 'insulin treated') and duration of T2DM (years). Cardiovascular risk factors will also be extracted, including diastolic blood pressure (mm Hg), systolic blood pressure (mm Hg), triglycerides (mmol/L), high-density lipoprotein cholesterol (mmol/L), low-density lipoprotein cholesterol (mmol/L), total cholesterol (mmol/L), creatinine, weight (kg), waist circumference (cm) and body mass index (kg/m$^2$). The presence of long-term conditions, including asthma, cancer, chronic kidney disease, chronic obstructive pulmonary disease, dementia, depression, diabetes, heart failure, hypertension, mental health, obesity, peripheral artery disease, chronic heart disease, stroke and transient ischaemic attack, and ischaemic heart disease, will also be extracted. Search terms will be based on the set of Read codes used for the quality and outcomes framework, a pay-for-performance system used within primary care in England.

Healthcare service utilisation variables will include the number of contacts with primary care (number of contacts with a GP practice), secondary care (number of appointments with specialists in hospital-based settings, including outpatient, inpatient elective, inpatient non-elective, Accident & Emergency) and mental healthcare during the timeframe of the study. The highly skewed nature of healthcare utilisation, with a few patients accounting for a large proportion of care, can complicate the modelling process.[29] Therefore, extreme high utilisers (top 1% for any utilisation variable) will be excluded.

## Quality checks and missing data handling

Data are unlikely to be missing at random[30]; therefore, no attempt will be made to impute numeric missing data, and continuous variables will be categorised with an additional 'missing' category included. Absence of clinical codes for diagnoses will be taken to indicate the diagnosis is not present. Clinically implausible values will be excluded with the Health Survey for England statistics used as a guide.[31]

## Bayesian networks

Machine-learning algorithms using BNs will be used to explore the main drivers of clinical deterioration in people with T2DM. A BN is a directed acyclic graph which combines both statistical and graph theory for representing conditional independencies.[32 33] In this acyclic graph, edges represent conditional dependencies; nodes that are not connected represent variables that are conditionally independent of each other. Each node is associated with a probability function that takes, as input, a particular set of values for the node's parent variables, and gives (as output) the probability (or probability distribution, if applicable) of the variable represented by the node.[32 33]

For example, if we consider that two variables (A, B) can affect a third one (C); and that A has a direct effect on B, the situation can be modelled as a BN. Each variable has two possible values ('true' or 'false'). The joint probability function is:

$$\Pr(C, A, B) = \Pr(C|A, B) \Pr(A|B) \Pr(B)$$

The model can be used to answer questions about the presence of a cause given the presence of an effect (ie, 'inverse probability'), such as what is the probability of C, given that A is true:

$$\Pr(C = T|A = T) = \frac{\Pr(A=T, C=T)}{\Pr(A=T)} = \frac{\Sigma A \epsilon \{T,F\} \Pr(C=T, A, B=T)}{\Sigma A,B \epsilon \{T,F\} \Pr(C=T, A, B)}$$

Obtaining a BN from data is typically performed as a two-step process. The first step is to determine the graph G, which contains the conditional independencies of the data ('structure learning'). Tree Augmented Naïve Bayes (TAN) is a semi-naive Bayesian Learning method. It relaxes the naïve Bayes attribute independence assumption by employing a tree structure, in which each attribute only depends on the class and one other attribute. A maximum weighted spanning tree that maximises the likelihood of the training data is used to perform classification. The second step is called 'parameter learning' and studies the probability distribution table of each node under the condition of knowing the BN structure essentially. Additional statistical description of BN can be found elsewhere.[32 33]

## Statistical analyses

Prognostic models will be defined using multi-dependence BN, such as the TAN classifier model, built over the set of available variables. The cohort will be randomly split into two groups, stratified by outcome. The first group, comprising 80% of the patients (n=1760), will be used to define the prognostic models (derivation cohort). Model parameters will be internally validated by comparing the area under the receiver operating characteristic (ROC) curve (AUC) in the derivation cohort with those calculated from a leave-one-out and a 10 times twofold cross-validation. The remaining 20% of patients (n=240) will be used as a validation cohort. The models generated will be visualised as risk matrices, using selected variables from logistic regression. To assess the discriminative ability of the risk matrices for each outcome, specific cut-off values will be chosen after performing an ROC analysis of the derivation cohort. The derived decision rules will then be evaluated, estimating sensitivity, specificity, accuracy, predictive values, likelihood ratios, post-test odds and prognostic ORs.

## Patient and public involvement

Patient partners will be included in the interpretation of our results, in the co-development of a dissemination strategy, and in summarising the research findings into lay summaries and reports, in order to raise awareness and stimulate public participation on this topic.

## Ethics and dissemination

This project has been approved by the Information Governance Committee at the WSIC (21 May 2020).

## EXPECTED RESULTS

We will develop and validate prognostic models for clinical deterioration in T2D. We will develop risk matrices that can be used to accurately predict the likelihood of a patient with T2D facing progressive disease, and can be used to timely identify high-risk patients, and deliver targeted early intervention. This work will also deliver the prototype of a prognostic predictive support system based on web forms.

## DISCUSSION

Use of EHRs and machine-learning methods have created an enormous opportunity for further refinement of risk prediction tools in the context of T2DM. Our study aims to provide valuable advances on this topic, by developing and validating a machine-learning tool able to predict the risk of clinical deterioration based on a comprehensive set of patient characteristics and using linked EHRs.

A previous study by Battineni *et al* conducted experiments to predict diabetes in Pima Indian females with particular machine learning (ML) classifiers.[23] However, this study used a much smaller sample (n=768 female patients), and used a dataset comprising only eight risk factors (age, and diabetes-related clinical factors). No

sociodemographic factors were included in the model. Also, this study used a dataset owned by the National Institute of Diabetes and Digestive and Kidney Diseases (the Pima Indian diabetes dataset), rather than routinely collected data, which are recognised to have a better performance for these models.[13] The same limitations apply to a study by Kahramanli *et al*, that used the same dataset.[24] This study is likely to have a large sample size, since it will use WSIC, which covers over 2.4 million patients in North West London. Additionally, data will be representative of 'real-life' patients as will be collected as part of routine clinical care. While previous evidence has shown that prediction models using EHRs data have better predictive performance than those using administrative data, it has also found that most of the models examined lacked inclusion of socioeconomic features.[13] In this study, patient-level linked data (comprising information from primary, secondary, social and community care) will allow us to explore the contribution of a comprehensive set of characteristics, including sociodemographic, geographic, clinical, patient activation and service utilisation.

This rich dataset will be used to develop a tool able to differentiate patients with T2DM based on their probability of facing clinical deterioration, using BNs, a machine-learning method particularly useful to explore the influences between variables. Previous literature comparing the prediction accuracies obtained by conventional statistical regression methods and machine-learning methods showed higher classification accuracies for machine-learning models, as measured by the AUC scores (65%–85% vs 80.8%–99.4%).[15] Compared with regression-based approaches, BNs present several additional advantages, including the generation of network structures in which relationships between variables can be easily communicated, their ability to apply Bayes' theorem to conduct individual-level risk estimation, and their easy transformation into decision models.[16] Additionally, BNs allow us to proceed with inference even in the presence of missing observations, representing an advantage in comparison with other ML methods.

Limitations of the study include the potential lack of accuracy of diagnosis of T2DM and potentially related risk factors and/or clinical deterioration outcomes; however, since T2DM is continuously monitored as part of the quality and outcomes framework,[34] it is an area where healthcare professionals are particularly incentivised to keep information completed and updated. For outcomes of clinical deterioration, there is a considerable redundancy of codes (ie, several codes may be used to describe the same event). To improve consistency, we have selected an extensive list of clinical codes which encompass symptoms and diagnoses to define these outcomes. Another limitation refers to the external validity of the tool, and therefore future work should consider replication in other populations.

With the increasing availability of EHRs, the growing awareness of the role of social determinants of health in health outcomes and the advances in machine-learning

methods, there is a growing opportunity to develop novel approaches for risk stratification. Using data-driven approaches, including novel factors, such as sociodemographic, ethnic, geographic factors, as well as the degree of patients' self-management ability or the patterns of healthcare usage, can provide a more granular picture of the patients that are more likely to deteriorate, and what are the main drivers for that deterioration. To identify these patients is of crucial importance for patient-centred population health and integrated care pathway delivery, and will allow for timely intervention, and more effective healthcare delivery.

**Contributors** ALN, PPR, AM, BG, TW, AD and EM contributed to the conception and design of this work. ALN wrote the manuscript with input from all authors. All authors approved the version submitted for publication.

**Funding** This work was supported by the National Institute for Health Research (NIHR) Imperial Patient Safety Translation Research Centre and by the PARSUK/FCT Bilateral Research Fund. Infrastructure support was provided by the NIHR Imperial Biomedical Research Centre (BRC). The research was enabled by the Imperial Clinical Analytics Research and Evaluation (iCARE) environment and used the iCARE team and data resources.

**Disclaimer** The views of the authors do not necessarily reflect those of the NHS, NIHR or the Department of Health.

**Competing interests** None declared.

**Patient and public involvement** Patients and/or the public were involved in the design, or conduct, or reporting, or dissemination plans of this research. Refer to the Methods section for further details.

**Patient consent for publication** Not required.

**Provenance and peer review** Not commissioned; externally peer reviewed.

**Data availability statement** No data are available. Patient-level data will not be made available.

**ORCID iD**
Ana Luisa Neves http://orcid.org/0000-0002-7107-7211

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
