## [Reviewer comments · BMJ Open]

ARTICLE DETAILS

TITLE (PROVISIONAL)	Using electronic health records to develop and validate a machine learning tool to predict type 2 diabetes outcomes: a study protocol
AUTHORS	Neves, Ana Luisa; Pereira Rodrigues, Pedro; Mulla, Abdulrahim; Glampson, Ben; Willis, Tony; Darzi, Ara; Mayer, Erik

VERSION 1 – REVIEW

REVIEWER	XIONG, Xiao-lu Nanjing Medical University
REVIEW RETURNED	15-Jan-2021

GENERAL COMMENTS	The reviewer completed the checklist but made no further comments.
--

REVIEWER	Battineni, Gopi University of Camerino
REVIEW RETURNED	26-Jan-2021

GENERAL COMMENTS	Authors presented an interesting study to develop a protocol to validate an ML tool in type 2 diabetes prediction. Although the study is interesting, it had some flaws before it needs to be accepted. ⊖ The abstract should be expanded as what is the study objective, how they design, sample size, and clear mentioning of preliminary outcomes rather than wind up in two subsections⊖ The introduction section has well described including existed literature. I would like to suggest add or compare the extensive studies on conventional ML models with deep learning models presented in [23]. Please identify some studies } Battineni G, Sagaro GG, Nalini C, Amenta F, Tayebati SK. Comparative Machine-Learning Approach: A Follow-Up Study on Type 2 Diabetes Predictions by Cross-Validation Methods. Machines. 2019; 7(4):74. https://doi.org/10.3390/machines7040074 } Humar Kahramanli, Novruz Allahverdi, Design of a hybrid system for the diabetes and heart diseases, Expert Systems with Applications, Volume 35, Issues 1–2, 2008, Pages 82–89, https://doi.org/10.1016/j.eswa.2007.06.004. ⊖ In the methods section, PAM is validated with 13 questions. the questionnaire administered is tailor-made, right? Does it contain parts of validated questionnaires? Please provide detailed information's
--

- ⊞ I am wondering that the results section is missing from the source file. I did not find any preliminary outcomes of the presented protocol
- ⊞ Since it has 2.4 Million sample size which is a relatively high number, how authors maintained this information in long periods. How about patient data (EHR) security?
- ⊞ Authors mentioned that this study aims to produce a comprehensive ML tool that links with EHRs? Could you elaborate more on this
- ⊞ Discussion section needs to be modified by including other studies that similarly related or perform possible comparisons
- ⊞ How this proposed ML tool could be different from existed ones

VERSION 1 – AUTHOR RESPONSE

Reviewer 1: Dr. Xiao-lu XIONG, Nanjing Medical University
Comments to the Author: no comments

Reviewer 2: Dr. Gopi Battineni, University of Camerino

Comment 1: Authors presented an interesting study to develop a protocol to validate an ML tool in type 2 diabetes prediction. Although the study is interesting, it had some flaws before it needs to be accepted.

Response: We thank the reviewer for the constructive comments. The suggestions have been incorporated, as described in the point-by-point answer below.

Comment 2: The abstract should be expanded as what is the study objective, how they design, sample size, and clear mentioning of preliminary outcomes, rather than wind up in two subsections.

Response: Thank you for this suggestion. As recommended, we have now re-structured the abstract and expanded it to the following sections: “Introduction”, “Objective”, “Sample and design” and “Preliminary outcomes”. As our approach is data-driven (i.e., we are using existing data to generate and validate new models, rather than finding evidence of a particular hypothesis), no specific sample calculations were applicable, and therefore not performed. This information is also added in the manuscript as part of the methods section: “The data used in the study are managed as part of Whole Systems Integrated Care (WSIC) systems, run and managed by the National Health System (NHS) and used for both direct care and approved research. This system pulls information from healthcare provider systems (local EHRs) under data controllers using a common model. Data are held in a secure NHS managed environment. As our approach is data-driven (i.e., we are using existing data to generate and validate new models, rather than finding evidence of a particular hypothesis), no specific sample size calculations were applicable, and therefore not performed.”

Comment 3: The introduction section has well described existing literature. I would like to suggest add or compare the extensive studies on conventional ML models with deep learning models presented in [23].

Response: Many thanks for this useful comment. We have now added two additional studies using deep learning neural networks and comparative machine-learning approaches.

The edited version of the manuscript reads as follows: “Recently, a few studies developed promising approaches applying deep learning neural networks and comparative machine-learning approaches for predicting T2DM [23,24,25].”

Comment 4: In the methods section, PAM is validated with 13 questions. The questionnaire administered is tailor-made, right? Does it contain parts of validated questionnaires? Please provide detailed information.

Response: Thank you for raising this important point. The PAM questionnaire administered to patients was the complete validated version, including its 13-questions. This information has been clarified in the manuscript, currently reading has follows: “The PAM is a validated tool of 13 questions, and it was delivered in its totality, without any edits or changes to the validated version [27]. Answers were weighted and combined to provide a score on a scale from 0 to 100 [27].”

Comment 5: I am wondering that the results section is missing from the source file. I did not find any preliminary outcomes of the presented protocol.

Response: Thank you for this important comment. We have now added a paragraph on “expected results”. The edited version of the manuscript reads as follows: “We will develop and validate prognostic models for clinical deterioration in T2D. We will develop risk matrices that can be used to accurately predict the likelihood of a patient with T2D facing progressive disease, and can be used to timely identify high-risk patients, and deliver targeted early intervention. This work will also deliver the prototype of a prognostic predictive support system based on web forms.”

Comment 6: Since it has 2.4 million sample size which is a relatively high number, how authors maintained this information in long periods. How about patient data (EHR) security?

Response: We thank the reviewer for raising this important point. The data used in the study are managed as part of Whole Systems Integrated Care (WSIC) systems, run and managed by the National Health System (NHS) and used for both direct care and approved research. This system pulls information from healthcare provider systems (local EHRs) under data controllers using a common model. Data are held in a secure NHS managed environment. (Full information can be obtained at <https://westlondon.nhs.uk/patients-and-carers/your-records/integrated-care-record/>). We have now added this information in the manuscript. The edited section of the manuscript now reads as follows:

“Data source and data management

Anonymised electronic health care records were accessed in the Whole System Integrated Care (WSIC) database [24]. Over 360 GP practices, 10 acute and specialist hospitals, 8 social care organisations and 2 mental health trusts and 2 community health trusts contribute to WSIC, which covers over 2.4 million patients in North West London, representing 30.0% of the London population [24]. Data from primary, secondary, community, social, and mental health care are linked at patient level. The data used in the study are managed as part of Whole Systems Integrated Care (WSIC) systems, run and managed by the National Health System (NHS) and used for both direct care and approved research. This system pulls information from healthcare provider systems (local EHRs) under data controllers using a common model. Data are held in a secure NHS managed environment.”

Comment 7: Authors mentioned that this study aims to produce a comprehensive ML tool that links with EHRs? Could you elaborate more on this?

Response: Thank you for highlighting this. The aim of this work is to design and validate a machine learning-based tool to identify patients with T2DM at high risk of clinical deterioration, based on a comprehensive set of patient level characteristics retrieved from a population health linked dataset. By ‘population health linked dataset’ we mean that these electronic records include data from primary, secondary, community, social, and mental health, linked at patient level. This information is described in the methods (Data source and data management), that reads as follows: “Anonymised electronic health care records were accessed in the Whole System Integrated Care (WSIC) database [24]. Over 360 GP

practices, 10 acute and specialist hospitals, 8 social care organisations and 2 mental health trusts and 2 community health trusts contribute to WSIC, which covers over 2.4 million patients in North West London, representing 30.0% of the London population [24]. Data from primary, secondary, community, social, and mental health care are linked at patient level.”

Comment 8: Discussion section needs to be modified by including other studies that are similarly related or perform possible comparisons.

Response: Thank you for this useful comment - we have edited the manuscript accordingly, adding a section in the discussions where we include other studies that perform similar comparisons. The edited version of the manuscript currently reads as follows: “A previous study by Battineni G et al (2019) conducted experiments to predict diabetes in Pima Indian females with particular ML classifiers. However, this study used a much smaller sample (n= 768 female patients), and used a dataset comprising only 8 risk factors (age, and diabetes-related clinical factors). No socio-demographic factors were included in the model. Also, this study used a dataset owned by the National Institute of Diabetes and Digestive and Kidney Diseases (the Pima Indian diabetes dataset [PIDD]), rather than routinely collected data, which are recognised to have a better performance for these models [14]. The same limitations apply to a study by Kahramanli H et al (2008), that used the same dataset.”

Comment 9: How this proposed ML tool could be different from existing ones?

Response: Thank you for this useful comment. This proposed ML tool combines four advantages:

- 1) Uses patient-level linked data (comprising information from primary, secondary, social and community care) will allow us to explore the contribution of a comprehensive set of characteristics, including socio-demographic characteristics (not included in most of models developed to date).
- 2) Uses a large dataset using linked EHRs (covering over 2.4 million patients in North West London).
- 3) Data are representative of ‘real-life’ patients as will be collected as part of routine clinical care. These data have shown to have a better predictive performance than administrative data.
- 4) Capitalises on these characteristics to develop a ML tool based on Bayesian Networks, a machine learning method particularly useful to explore the influences between variables. Additionally, Bayesian Networks allow us to proceed with inference even in the presence of missing observations, representing an advantage in comparison with other ML methods. These aspects are highlighted as part of the ‘Discussion’ section. The edited version of the manuscript currently reads as follows: “This rich dataset will be used to develop a tool able to differentiate T2DM patients based on their probability of facing clinical deterioration, using Bayesian Networks, a machine learning method particularly useful to explore the influences between variables. Previous literature comparing the prediction accuracies obtained by conventional statistical regression methods and machine learning methods showed higher classification accuracies for machine learning models, as measured by the AUC scores (65-85% vs 80.8-99.4%) [16]. Compared to regression-based approaches, BNs present several additional advantages, including the generation of network structures in which relationships between variables can be easily communicated, their ability to apply Bayes’ theorem to conduct individual-level risk estimation, and their easy transformation into decision models [17]. Additionally, BNs allow us to proceed with inference even in the presence of missing observations, representing an advantage in comparison with other ML methods.”

REVIEWER	Battineni, Gopi University of Camerino
REVIEW RETURNED	21-Apr-2021
GENERAL COMMENTS	The authors successfully addressed the raised comments. I recommend to accept this work as in same manner.